# Efficiency of Obstetric Services in Germany—The Role of Variation and Overheads

**DOI:** 10.3390/healthcare12010009

**Published:** 2023-12-19

**Authors:** Steffen Flessa

**Affiliations:** Department of Health Care Management, Faculty of Law and Economics, University of Greifswald, 17487 Greifswald, Germany; steffen.flessa@uni-greifswald.de; Tel.: +49-3834-420-24-77

**Keywords:** obstetrics, small hospitals, efficiency, economies of scale, delivery service, Germany

## Abstract

The number of obstetric departments in German hospitals has declined in the last decades. In particular, rural hospitals are challenged to sustain their delivery services. In this paper, we analyse the role of variation and overheads of obstetric departments from the perspective of current and future German hospital financing. For this purpose, we develop a Monte Carlo simulation model that analyses the workload of the labour room and the obstetric ward. The results show that a hospital with less than 640 deliveries per year cannot break even. In order to offer services 24 h per day, 365 days per year, five nurses, five midwives, and five gynaecologists are needed. This results in high fixed costs. At the same time, the variation coefficient of the labour room and the obstetric ward declines with an increasing number of deliveries. Consequently, small hospitals have a higher risk of over- and under-utilization in the course of the year. This paper acknowledges that economics is not the only decision dimension. The quality of the institution and the transport to the hospital have to be considered, as well as the population’s wish for nearby services. However, the simulations clearly demonstrate that unless the hospital financing system is changed fundamentally, the decline in the number of hospitals offering delivery services will continue.

## 1. Introduction

In 2003, German healthcare faced a reform of the hospital financing system with a strong impact on the entire sector. The shift from daily rates to flat rates based on diagnosis-related groups (DRGs) was made in order to improve the efficiency of each hospital and of the entire healthcare service in Germany [1]. One major objective was to reduce the number of small hospitals, which were seen as highly inefficient, with high overheads and high cost per case [2]. Consequently, it was perceived that a concentration on fewer hospitals with more beds per hospital would be preferable [3]. As a consequence of the reform, the number of hospitals in acute care declined from 2221 (31 December 2002) to 1893 (31 December 2022), while the number of beds per hospital increased from 244 to 254 in the same period [4].

While the decline of 13% in the number of hospitals might not seem impressive in a period of almost 20 years, many small hospitals are even smaller today and have lost complete departments. In many cases, closures of hospitals and/or departments were accompanied by strong resistance from the local population, in particular in rural areas [5]. The population sees the existence of hospital services in their areas as essential, but many of them cannot survive with the existing DRG rebates [6]. Many institutions attempt to break even by offering well-paid specialised services with the risk of compromising the quality of services in these smaller units. Consequently, the ongoing hospital sector reform will abolish the original DRG system for smaller hospitals (in particular in rural areas) and prepare an exclusive list of elementary services they are permitted to offer [7,8].

Obstetric hospital services are a major focus point of the ongoing discussion about the future of small rural hospitals in Germany. On the one hand, the population insists on nearby accessible hospitals for delivery and paediatrics, while the costs and quality of services call for concentration on fewer centres. One underlying problem is the strong decline in the number of deliveries. While this statistic was more than a million in 1970 and still more than 900,000 at reunification in 1990, it declined to little more than 660,000 in 2011. It reached another local maximum in 2021 with 795,492 newborns, but in 2022, the statistic declined to 738,819. As many as 96.25% of children were born in hospitals, but in particular, rural regions have, by far, fewer deliveries than in former times [9].

At the same time, the number of hospitals offering obstetric services has declined. In 1992, 1186 hospitals offered these services; in 2007, 865 hospitals, and in 2021, merely 621 [10] hospitals. In particular, small obstetric departments had to close down. In Germany, an obstetric department is considered “small” if it has less than 500 deliveries p.a. In 2010, there were still 260 departments of that size, and in 2021, merely 138 [10]. Figure 1 shows the number of deliveries in obstetric departments in 2021 [11]. The average obstetric department had 1071 deliveries with a huge range between 1 (where the department was closed that year) and 5662. A total of 67 hospitals had less than one delivery per day, which was 207, which is less than 640 per year.

An example is the so-called “island hospitals” on the northwest German islands of Föhr, Sylt, and Helgoland. When the German DRGs (G-DRG) were introduced, these islands had obstetric wards in small hospitals. In 2003, there were eight deliveries on Helgoland, but the department was closed in 2004. There is still a mini-hospital with 24 beds on the island. The hospital in Westerland on Sylt closed the obstetric services in 2013, having seen some 80–100 deliveries per year. A small hospital of 84 beds for general services is left. Finally, the delivery services in Wyk on Föhr Island were closed in 2015. In 2014, 64 deliveries were still performed on this island. Since then, mothers went to Niebüll Hospital, a smaller hospital on the mainland with 135 beds and 236 deliveries in 2015, but the respective obstetric department was closed in 2016, so mothers have to deliver at Husum Hospital (770 deliveries in 2022). It is accepted that this is too far away for delivery, so the future mothers have to leave their islands and come to Husum Hospital 14 days before the expected delivery date in a special boarding refinanced by the statutory health insurance fund.

However, even today, the number of deliveries in small and mini-hospitals is considerable. The German DRG system distinguishes four cases of non-surgical deliveries:O60A: Vaginal delivery with several complicating diagnoses, at least one severe, duration ≤33 weeks or with complicating constellation;O60B: Vaginal delivery with several complicating diagnoses, at least one severe, >33 weeks, without complicating constellation or tamponing of bleeding or thromboembolism during gestation period without surgical procedure or severe or rather severe complicating diagnosis ≤33 weeks;O60C: Vaginal delivery with severe or rather severe complicating diagnosis or duration ≤33 weeks;O60D: Vaginal delivery without complicated diagnoses, >33 weeks.

Table 1 shows that the vast majority of deliveries are the “normal deliveries” of O60D. The more complicated a delivery, the less frequently it occurs. The table also shows that small hospitals focus a lot on “normal deliveries”, while more complicated cases are usually transferred to higher-level and bigger hospitals. The same could be shown for Caesarean Sections (CS) with DRG O01A-O01H (226,677 CS in 2023, 6% in hospitals with less than 200 beds) and DRG O02A-O02B (delivery with complicated surgery; 18,663 cases, primarily in major hospitals) [12]. The DRG is calculated per adult case, i.e., one DRG, irrespective of whether a mother delivers one child, twins, or more children. Consequently, the number of newborn children is higher than the number of deliveries recorded in the DRG statistics. A total of 12,532 deliveries resulted in more than one child in 2022 [13]. Furthermore, 14,401 children were not delivered in hospitals [14]. In this paper, we will focus on the “normal simple” delivery of DRG O60D as the most relevant delivery procedure in smaller hospitals. For 2022, Table 1 shows 396 cases in hospitals between 0–49 beds, 70 between 50–99 beds, 7897 between 100–149 beds, and 18,207 between 150–199 beds. The biggest number of deliveries is performed in hospitals with 300–399 beds, and 55% of all deliveries are performed in hospitals with less than 500 beds.

Hospitals with obstetric services are unequally distributed in Germany, and accessibility is diverse [16]. In particular, remote districts (such as northwest and northeast) have areas where mothers have to travel more than 40 min to reach a hospital with obstetric services—a distance that is seen as unacceptable by many and which leads to fierce protests against further closure of institutions. While 95% of German mothers in urban settings had to travel less than 20 min, only 74% of mothers in rural regions could reach an obstetric service within that time. In the state of Brandenburg, the respective figure was 68%, and in Mecklenburg-Vorpommern, it was only 58% (2016) [16]. Closing down more obstetric departments would increase the inequity of delivery services.

The reasons for closing down obstetric services are manifold. Frequently, it is difficult to find a sufficient number of midwives, nurses, and gynaecologists willing to work in a remote place with a low workload. At the same time, small hospitals have problems breaking even. Likewise, a small number of deliveries inclines with a higher mortality risk for mother and baby in case of a rare emergency. However, evidence is more anecdotal for Germany, and there is truly little quantification of the efficiency of obstetric services in Germany. Consequently, evidence-based decision-making becomes almost impossible. There is a major need to understand the economies of scale of obstetric departments in order to make wise decisions on subsidy or closure of obstetric departments in German hospitals.

In this paper, we would like to demonstrate that smaller obstetric departments have disadvantages in comparison to bigger departments. The handicap is a consequence of fixed costs and of the deviation of the arrival rates and length of stay. For this purpose, the next section describes a simulation model to calculate the consequences of different workloads of obstetric units. Afterwards, we show the results and discuss the consequences. This paper closes with some conclusions.

## 2. Methods

An obstetric department can be seen as a waiting system where patients arrive, have a delivery in the labour room and stay in the obstetric ward for a number of days [17]. The number of arrivals and the length of stay are random processes. Figure 2 indicates the waiting system with a waiting room before the labour room, a first service channel (labour room), a second waiting area (ward) and a second service channel (discharge).

In the following, we assume that the number of arrivals (deliveries) is Poisson distributed with an arrival rate λ [18]. We assume that the waiting area is indefinite, i.e., no patients are rejected. The service time in the labour room is assumed to follow a negative exponential distribution with a service rate of μ [18]. The length of stay in the ward is assumed to have a log-normal distribution with an expected value of *l* and a standard deviation of σ.

Consequently, we develop a queuing model with log-normal service time, parallel service channels and two subsequent services. It can be denoted as M/G/k*2 ∞/FIFO. The first “M” stands for a Markov-like arrival process, as the Poisson distribution is exactly a Markov process. “G” denotes a general distribution of the service process, in this case, a log-normal service time distribution. “k” stands for the number of parallel labour rooms, and “*2” indicates that there are two subsequent service departments (labour room and obstetric ward). All patients will be accepted (∞), and the first patients will receive first services (FIFO). Contrary to other authors [19,20], this model does not develop a traditional Markov model as the focus of this paper is on the variation, not the average in the steady state equilibrium [21].

The main source of data is the G-DRG-browser [12], offering detailed information on all G-DRGs. In the following, we will concentrate on the DRG O60D (vaginal delivery without complicated diagnoses; >33 weeks) as this is the most frequent DRG coded by smaller, rural hospitals. In 2022, there were 358,532 deliveries recorded under DRG O60D. It is the most frequent obstetric DRG in smaller hospitals.

The average length of stay for this DRG is 2.9 days, with a standard deviation of 0.8 days. If we assume that *d* is the number of deliveries per year, the arrival rate per day can be calculated as λ = *d*/365. Table 2 shows the respective constants of the basic simulation. Table 3 indicates this model’s variables.

The number of staff members of category *i* must suffice to offer services 24 h per day, 365 days per year (24/7). Assuming that a member of staff works 46 weeks per year (52 weeks minus regular and sickness leave, further training, etc.), 40 h/week, the minimum number of staff per category (*M_i_*) is 5 (rounding-up of 365·24/(40·46)); i.e., the hospital needs five nurses, midwives and doctors to offer their services for 24 h per day throughout the year [22]. This figure is the minimum number of staff in a hospital. If the number of deliveries and occupied beds is higher, the capacity must be increased accordingly. With the resource consumption of *R_n_*, the number of beds (*B*) and the number of deliveries *D*, the number of staff required can be calculated as
$$Nurses:\;X_{n}=max\left\{\frac{R_{n}\cdot B}{46\cdot 40};5\right\}$$
$${Mid-wives:\;X}_{m}=max\left\{\frac{R_{m}\cdot D}{46\cdot 40};5\right\}$$
$${Gynaecologists:\;X}_{d}=max\left\{\frac{R_{n}\cdot D}{46\cdot 40};5\right\}$$

The number of beds (*B*) should suffice in 85% of days (security level *p_w_*); i.e., *B* is the 85%-quintile of the patients throughout the year. Likewise, the labour room capacity (for doctors and nurses) should be appropriate in 85% of days; i.e., *D* is the quintile of the number of deliveries per day with *p_l_*%.

This model simulates the number of deliveries (*D_t_*) and the number of patients in the obstetric ward (*P_t_*) for each day t (t = 1.365). As the focus of this paper is on small, rural hospitals, we do not consider complicated deliveries and Caesarean sections. Based on this simulation, this model calculates the number of staff required to comply with a service readiness of the labour room and the obstetric ward of *p_n_*, *p_m_*, and *p_w_* for nurses in the obstetric ward, midwives in the labour room, and gynaecologists in the labour room/ward, which is again the basis for all further calculations.

To assess the consequences of variation in parameters on the economic situation of a hospital, it is necessary to know the costs and revenues. The German DRG Institute (InEK) calculates the average unit cost per DRG based on an annual sample. Table 4 exhibits the cost for one case of DRG O60D for different cost categories and cost centres according to InEK (browser of 2023). Since 2019, the cost of direct nursing (normal ward, ICU) has been excluded from the adjusted G-DRG system (aG-DRG) and calculated separately as a daily rate. Consequently, the respective amount was computed under the assumption of an average length of stay of *l* days, a nursing cost weight of *c_n_* per day, and a value of EUR *v_n_* per cost weight. Thus, we calculate the cost of direct nursing in the table as *l∙c_n_∙v_n_*; i.e., EUR 2.9∙0.6702∙230 = EUR 447.02 per case.

Based on this matrix, we can calculate the total cost, the marginal contribution, and the profit/loss of the department. We assume that the overheads for medical and non-medical infrastructures (column 7 in Table 4) are constant per case; i.e., economies of scale are only harvested within the obstetric department, not within the general hospital administration. Thus, we receive
R=D·r
$$C=X_{n}\cdot S_{n}+X_{m}\cdot S_{m}+X_{d}\cdot S_{d}+D\cdot \left(v_{d}+v_{o}\right)$$
$$m=R-X_{n}\cdot S_{n}+X_{m}\cdot S_{m}+X_{d}\cdot S_{d}+D\cdot v_{d}$$
π=R−C

This model can be described as a time-series Monte Carlo simulation [23] and was implemented in Delphi XE. For each day (t = 1.365), the number of arrivals, deliveries, and length of stay are chosen as a random number. The delivery room and the beds in the ward are occupied accordingly. Patients are discharged after their randomly chosen length of stay.

## 3. Results

### 3.1. Basic Simulation

Figure 3 exhibits the number of patients in the obstetric ward per day of simulation.

This simulation generated 514 deliveries. On average, there were 1.41 deliveries per day (standard deviation 0.36); the maximum number of deliveries per day was five, and the minimum was zero. The most frequent number of deliveries per day was one. Consequently, we had 4.07 patients in the ward (standard deviation of 1.82) with a range between zero and ten. The most frequent number of women in the obstetric department was three. The department with four beds had an occupancy rate of 101.71%, while the department was over-occupied 39.18% of the days. After increasing the capacity to five beds, it had an occupancy rate of 81.37% with an over-occupancy on 24.11% of days. For six beds, the occupancy rate is 67.81% with 7.95% of days over-occupancy; i.e., at a security level of 85% (*P_w_*), we would require six beds.

Assuming that the number of staff of each category should be sufficient in 85% of days (*p_n_*, *p_m_*, *p_w_*), we would require 2.4 nurses, 4.2 midwives, and 0.60 gynaecologists to cover the workload. However, as the minimum number to offer 24/7 services is five for each category, the respective number of staff is five for each cadre.

Based on this calculation, the hospital has direct costs of EUR 1,002,604.68 for the 514 deliveries. The highest costs are the salaries for gynaecologists (EUR 450,000.00), nurses (EUR 225,000.00), and midwives (EUR 225,000.00). Pharmaceuticals and materials are negligible, but an amount of EUR 358,571.54 is allotted as general medical and non-medical overhead from the hospital administration, so the total full cost is EUR 1,361,176.22. The 514 deliveries generate a revenue of EUR 1,184,016.17, so the marginal contribution of the obstetric department is EUR 181,411.49, with a loss of EUR 177,160.05. For more details, see Table A1 in the Appendix A.

In summary, we can say that 138 German hospitals have less than 500 deliveries per year and have a deficit with this service. Midwives and doctors have less than two deliveries per week; i.e., midwives only spend 34.60% of their working time directly engaged in patient services, while doctors spend 4.94%. Consequently, the occupancy of the staff required to provide 24/7 services is very low, resulting in high unit costs.

### 3.2. Scenarios

In the following, we would like to discuss some scenarios in order to determine the underlying causes of the inefficiency of obstetric departments. Firstly, we would like to analyse the impact of different workloads and determine the break-even point for obstetric departments under the assumption of the German DRG system.

Figure 4 exhibits the marginal contribution for different numbers of deliveries per year. It must be noted that a stochastic simulation will not necessarily produce exactly the number of deliveries expected, as it is a random process. However, the tendency is obvious.

With the aG-DRG system of 2023, an obstetric department can break even with 640 deliveries per year. On average, it has 1.92 deliveries per day (mode = 2) and 5.56 patients in the ward. With a security level of 85%, we would require eight beds in the ward, five nurses, five midwives, and five gynaecologists. The ward would be over-occupied on 14% of days with an average occupancy rate of 72.02%. Midwives would be occupied 43.08% of their time, and gynaecologists 6.15%. If we assume that the obstetric department does not have to contribute to the coverage of medical and non-medical overheads, the break-even point would be 428 deliveries per year.

If we compare the statistics for different numbers of deliveries, it is obvious that the coefficient of variation for patients in the ward and for deliveries per day declines with the increasing number of deliveries per year. At the same time, the occupancy rate strongly increases. This rate is calculated based on the number of beds required to safeguard the security level of 85%. The relative variation represents the risk that strongly declines with workload increase. Small obstetric departments experience much more variation. To safeguard the security level, they require more “security capacity”, which leads to poorer utilization of services.

The number of staff required to safeguard the security level is constant for a wide range. The number of midwives is constant at five until about 740 deliveries per year. For nurses, it remains on this level until the workload exceeds 1200 deliveries p.a. Most striking is the number of gynaecologists. Even with 2500 deliveries per year, the five gynaecologists will not be fully occupied; i.e., an obstetric ward alone will not provide sufficient workload for the specialists. For more details, compare Table A2.

In the following, we will analyse the impact of changes in parameters representing different standards. The first analysis assumes that the costs of gynaecologists are not fixed but are the product of the number of deliveries and physician cost per delivery expressed in Table 4 (EUR 455.02 per case). This assumption is true for the following two cases:(1)Deliveries and medical services for deliveries are performed by non-gynaecologists. This could, for instance, mean that a small hospital does not hire gynaecologists but uses the services of the existing surgeons. In this case, the surgical department charges the obstetric department for the hours worked;(2)The hospital has a general gynaecological department with a sufficient number of specialists. In this case, the gynaecological department charges the obstetric department for the hours worked.

If we assume a standard hospital with 500 deliveries per year under these assumptions, the cost of medical personnel decreases from EUR 450,000 to EUR 233,800.28. Consequently, the hospital makes a profit of EUR 38,959.67. Further analysis indicates that the hospital must have at least 474 deliveries per year in order to break even if they use the medical services from other departments. For more details, see Table A3 in the Appendix A.

Furthermore, we analyse the impact of different security levels on relevant parameters. As stated before, the security level expresses the likelihood that a capacity (beds, nurses, midwives, gynaecologists) suffices. For instance, a security level of 85% means that a resource will not be overutilized in 85% of all days. Table 5 shows the consequences of different security levels for hospitals with about 500 and about 700 deliveries per year. It becomes obvious that a high-security level requires more beds and more staff. For a hospital that does not break even and which is grossly underutilized, a strict security level will not necessarily have an impact on resource requirements. Consequently, only unrealistic levels (e.g., P = 100%) have an impact on staffing (the number of midwives increases to 7) and profit (deficit increase from EUR 177,160.05 to EUR 267,160.05). The number of beds, however, strongly depends on the security level. Likewise, the occupancy rate and actual over-occupancy decrease with an increase in security level. The consequence of a strict security level is a poorer utilization rate of the services.

A hospital which breaks even (e.g., N = 700) demonstrates the same consequences of changing security levels. However, the number of staff changes earlier, and the hospital is at risk of not breaking even. If an institution wants to have sufficient capacity at any moment, it is likely to pay for this service availability with a deficit.

Finally, we can change the work intensity in order to demonstrate the consequences of quicker (and likely less patient-oriented) work. For this purpose, the capacity utilization per service unit is changed. Originally, we assumed two hours of nursing time per patient per day, 7 h of midwife time per delivery, and one hour of gynaecologist time per patient per day. Table 6 shows the consequences of altered time consumption on the respective parameters. For underutilized hospitals, it does not make sense to increase the working speed. The number of staff is always the minimum to upkeep services 24/7. Even increasing the time consumption per service unit does not have an influence on gynaecologists and nurses, as they are underutilized anyhow. For midwives, however, a strong increase in time per delivery will lead to a higher demand for midwives and a higher loss.

If a hospital is above the break-even point (e.g., N = 718), reducing the working time per service unit does not make a difference either. However, if the working time increases, it will have a major impact on the number of staff members. The number of midwives reacts earlier than the number of nurses, while the number of gynaecologists does not react at all. Increasing the speed of work will, however, have a major impact on hospitals with a high workload (here: N = 1495). It can reduce the number of staff (up to the minimum) and, thus, increase its profit.

## 4. Discussion

The German hospital industry has experienced a decline in the number of obstetric departments, particularly in small hospitals with less than 500 deliveries p.a. The simulation indicates that this is a consequence of overheads and variation. With the current hospital financing system in Germany, hospitals with less than 640 deliveries p.a. cannot break even. Even if we assume that the leadership of the hospital will not require that the obstetric department contribute to recovering the medical and non-medical overheads of the hospital, we will still require 428 deliveries p.a. to recover the departmental marginal cost. The main reason for that is that obstetric departments have to offer their services 24 h per day, 365 days a year. To do so, they need midwives and nurses in the hospital at any hour of the year. Some hospitals try to avoid this by offering delivery services only from 8 a.m. to 5 p.m., Monday to Friday. The consequence is a strong increase in “elective deliveries”, i.e., induced birth and/or Caesarean Sections. Both are not desirable from a quality perspective.

Another alternative to reduce costs is to use the services of other medical specialists who are already in the hospital but do not work for the obstetric department regularly. In this case, the break-even point declines to 474 deliveries p.a. This looks like an alternative for small, rural hospitals (particularly island hospitals), but this number is still, by far, higher than the respective statistics of these hospitals. For bigger hospitals with a full gynaecological department, it is expected that these specialists spend most of their time with other gynaecological patients. Consequently, the costs for the deliveries are comparatively small. Nevertheless, for small hospitals that only have an obstetric department but do not treat general gynaecological cases, it is almost impossible to break even while hiring specialists only for deliveries.

Furthermore, the costs of the entire system could be reduced by a number of changes in the mode of transport and delivery. For instance, helicopter services can overcome even long distances in a relatively short time so that access to delivery services becomes feasible within a few minutes. An emergency helicopter commutes with a speed of 200–250 km/h; i.e., even if we concentrate all deliveries only in major hospitals, the distance would no longer be a limitation. However, as Röper et al. demonstrate [24], an emergency helicopter costs about EUR 70 per minute of flight, which makes air transport quite expensive. At the same time, transport distance is not only a challenge for delivering the mother to the hospital but also for the relatives and friends wishing to visit her and for the mother and baby travelling back home after discharge.

Another alternative under discussion is strengthening the independent midwives and home delivery. Having a choice of child delivery either at home or in the hospital is perceived as a human right [25]. While there might be some pros and cons in central and urban areas, it is obvious that this is no solution for rural areas, as the small number of deliveries would not allow independent midwives to earn their living. At the same time, the time and distance to the next hospital would be too high in case of a complication.

The current debate on hospital financing is going in a new direction [7]. The fixed costs of departments of small hospitals, which are seen as essential for the services for the catchment population, should be covered by a fixed income (fixed budget), and merely the variable cost should be covered by the DRG. The separated nursing budget under the aG-DRGs (starting from 2019) is already going in that direction, but in the future, the highest costs could be recovered, irrespective of workload. This should guarantee the survival of smaller units. For the hospital with 500 deliveries, the fixed budget would be EUR 1,296,592.12 (personnel, allotted overheads), while the variable costs for pharmaceuticals and materials are EUR 125.65 per delivery. However, some also recommend that the fixed budget should cover only 60% of the planned cost at the planned number of deliveries, in this case, EUR 816,705.73 fixed budget with a variable income of EUR 1059.25 per delivery. Figure 5 shows the respective revenue functions. “Revenues” represents the current situation; “Revenues*” is the fixed revenue function under the assumption that 60% of the budget is the expected workload. Both functions coincide at N = 640, i.e., at the break-even point. The last function is “Revenue+”, which assumes that the total fixed costs are covered by a fixed budget, and only the variable costs are covered by variable income. This function is also close to the real cost function. Revenue* and Revenue+ coincide with the expected number of deliveries (N = 514).

If the number of deliveries is less than 514, “Revenue+” is best, and the current situation “Revenue” is the worst for the hospital. Between 514 and 740 deliveries, the proposed function “Revenue*” is best; i.e., for most small hospitals, the expected change in hospital financing would be supportive, but unless the entire fixed costs (“Revenue+”) are recovered, the risk of running at a loss is still high. It is better than before, but the proposed change to recovering 60% of the total cost at the expected workload by a fixed budget will not solve the problem of dying obstetric wards completely.

Currently, hospitals can only react by improving their technical efficiency. This can be achieved by reducing the consumption of materials per delivery, but the respective amount is rather small (EUR 125.65 per delivery for materials and pharmaceuticals). Furthermore, they can try to reduce staff by making them work “faster”, i.e., requiring less working time per delivery. As we saw, this increases the efficiency of larger obstetric departments but not smaller ones. Their main problem is the underutilization of staff required for 24/7 services. As they stand idle waiting for customers most of the time, it does not make much sense to foster them working faster.

The alternative would be reducing quality by altering security levels and using medical specialists from other departments. This model assumes that staff and beds should not be overutilized in more than 15% of days. This means, for instance, that floor cases are permitted on 54 days per year; otherwise, the number of beds has to be increased. For small hospitals, overutilization of staff is not a problem, so they cannot even reduce their economic pressure by allowing for more days of overutilization. As the number of staff is determined mainly by the minimum to provide 24/7 services, merely the number of beds reacts to the alteration of the security level, and the beds per se have very little marginal costs.

As shown above, using services from other medical specialists looks more promising. As we showed, the break-even point reacts to this decision. According to German law, an obstetric department requires specialized gynaecologists. Consequently, this alternative cannot be fully applied. It is possible that hospitals with small obstetric departments have a gynaecologist during regular working hours (8 a.m.–5 p.m. Monday–Friday) and use the services of the bigger surgical department any other time. This could be an alternative for the island and other rural hospitals. However, their number of deliveries is so low that even this will not allow them to break even.

Another problem is the variation in demand. While the impact of overheads on the break-even point was demonstrated before, the relevance of the variation in demand requires a stochastic simulation, which is more complex and not frequently performed. For instance, Markov models of hospital departments focus on averages and steady-states, so variation in demand can hardly be studied [19,20]. Another approach was chosen by van den Berg et al. [26], who developed a Linear Program (LP) to determine the most cost-effective mode of delivering obstetric and pediatric services in a region in Northeast Germany. Their model has the advantage that it is not simulative-experimenting (in the sense of ‘let us change some variables and see what happens’) but optimizing; i.e., this model calculates the number of patients treated in hospitals of different sizes that minimises total regional costs. However, LPs are deterministic models, which do not permit an analysis of variation.

Our analysis demonstrates that bigger units with many deliveries will have a smoother utilization of their capacities, i.e., the variation coefficient declines with increasing number of deliveries (theoretically with 1d) [27]. Smaller hospitals face the problem of sometimes being overcrowded and sometimes being ‘half-empty’. This requires a comparably higher staffing level, is frustrating for the members of staff, and calls for stochastic simulation models to analyse this variation.

The satisfaction of co-workers depends on the work intensity [28]. Most medical co-workers have a high intrinsic motivation and want to perform well. It is frustrating if they are underutilized. Only very few nurse-midwives would like to sit idle most of their working time waiting for rare customers to come. They would like to work in a place where they can do what they have learnt; this is why they selected this profession. Gross under-utilization of staff—as is common in small obstetric departments—is frustrating. In addition, it will constitute a major barrier in hiring professionals. The small, rural hospitals have major problems filling their positions.

This model demonstrates the structural disadvantages of small obstetric departments. However, this model does not consider the impact of the number of deliveries on the quality of services. The relationship between quality and quantity of deliveries is complex and beyond the scope of this paper, but several aspects have to be considered by decision- and policy-makers, as well as future research.

Learning effect: It is common sense that big centres with high quantities have advantages, and there is evidence that the mortality of mothers and children depends on the routine. The ‘normal’ delivery can be handled by hospital staff even if they have only one delivery per week. In the case of an unforeseen emergency, the routine will determine the risk for the mother and baby. As shown by other countries, the mortality declines significantly if more than 1000 deliveries are performed in one centre. In a recent meta-analysis, Walther et al. showed that there was a clear relationship between the volume and the outcomes of deliveries [29]. Similar studies are not known for Germany, but it is obvious that the reaction to complications depends on the routine;Rates of CS: It is likely that higher routine would also reduce the rate of Caesarean Sections, so CS rates should be lower in hospitals with more deliveries. However, these bigger institutions are usually also on a higher level with more severe cases. Both effects are difficult to discriminate, so some studies indicate that bigger hospitals have a higher CS rate [19];Assessment of risks: The assessment of the individual risk of complications is rather complex, particularly as a prognosis. A higher routine might increase the quality of the risk assessment; i.e., smaller hospitals might have a limitation in assessing the risks correctly. Furthermore, the best way to reduce the risk is to concentrate all deliveries in a place of high volume without assessing whether it might be feasible to do the delivery in a smaller hospital;Trade-off with long distances: The respective studies usually consider only the technical effectiveness in the hospital, not the entire service chain. Patients have to come to the hospital, return home, and receive further support afterwards. Concentrating all delivery services in hospitals with, for instance, more than 1500 deliveries per year might increase the resulting quality of this institution, but it might also lead to cases in which the patient cannot reach the hospital on time. Most likely, a delivery in a small hospital with professionals is still much better than a delivery in an emergency car;Will of the patient: The majority of mothers prefer a delivery that is near their place of residence. They want to reach ‘their’ institution; they ask for further support from the midwife; they want their relative to visit them, and they would like to see their place of residence in the passport of their children as their place of birth. Indeed, German law enforces that the place of delivery is entered as the place of birth in the birth certificate. When a hospital closes down the delivery services, the place of future births changes. For some mothers (e.g., on an island with a strong cultural feeling of ‘we the islanders’), this is a challenge.

Consequently, the assessment of the quality of deliveries is complex because we have to distinguish the technical quality of the institution versus the loss of quality due to long distances, the objective and subjective dimensions of quality, as well as the perception of risks. This calls for further research, particularly in Germany, where little is known about these dimensions of healthcare.

This paper is subject to a number of limitations. Firstly, the simulation model concentrates on DRG O60D. As shown in the introduction, we did not consider DRGs O60A-C and DRG O01A-O01H [12]. As shown in Table 1, small hospitals perform Caesarean Sections and more complex deliveries, particularly preterm labour. However, as the table indicates, these more complex diagnoses are less relevant for small hospitals. At the same time, all that we said about quality and cost challenges of small numbers for DRG O60D applies manifold to the other DRGs so that our arguments are even strengthened once they are included in the discussion.

In addition, we ignored DRGs O01A-O01F (Caesarean Sections) and O02A-O02B (vaginal deliveries with other surgical procedures). However, DRG O60D is the most frequent normal delivery (about 78% of deliveries). Hospitals with small obstetric wards have mainly this DRG as all risky and surgical deliveries are referred to bigger centres anyhow. Consequently, a complete analysis of the entire healthcare system would have to consider other DRGs as well, but for an analysis concentrating on smaller units, it is acceptable to focus on this DRG alone.

Another limitation is the assumption that the arrival of patients and the service in the labour room are all Markov processes, while the length of stay is log-normally distributed. This assumption is based on queueing theory [21], but we did not test the distribution based on empirical data. However, it is unlikely that the distribution of arrival and service processes, as well as the length of stay, would have a major impact on the insights flowing from this analysis.

We also have to state that several assumptions (e.g., security levels and resource consumption) are good estimates, which could not always be based on empirical data or the literature. Instead, we made some sensitivity analyses and scenarios.

## 5. Conclusions

Based on these findings, we can conclude that decision-making in obstetric departments in German hospitals should be evidence-based. Closing down and concentrating on the respective departments should be considered carefully in many dimensions. The economics of the hospital under the current financing scheme, the total cost for the society (incl. emergency transport, maternity waiting homes, etc.), the quality of the institution and transport, as well as the wish of the population for accessible, nearby services must be considered. In this paper, we contributed to the debate by demonstrating the impact of volume on cost and variation. This is one part of the puzzle, but more research is needed to include the other dimensions in the decision model and develop the ‘economics of childbirth’ as a standard [30]. Currently, more studies about economics, quality, and equity of obstetric care are known from low-income countries than from Germany [31].

Birth-giving and, consequently, obstetric services are highly sensitive topics. Much debate is going on in Germany on the number and distribution of respective hospital departments. This debate is not always well-informed about the consequences of sustaining or closing the services. The role of the economist and of decision-support systems might not necessarily be to make the decision but to shed light on alternatives and their consequences in many dimensions. Bringing transparency to the decision process and showing the consequences of selected alternatives is a major contribution health economics can offer.

## Figures and Tables

**Figure 1 healthcare-12-00009-f001:**
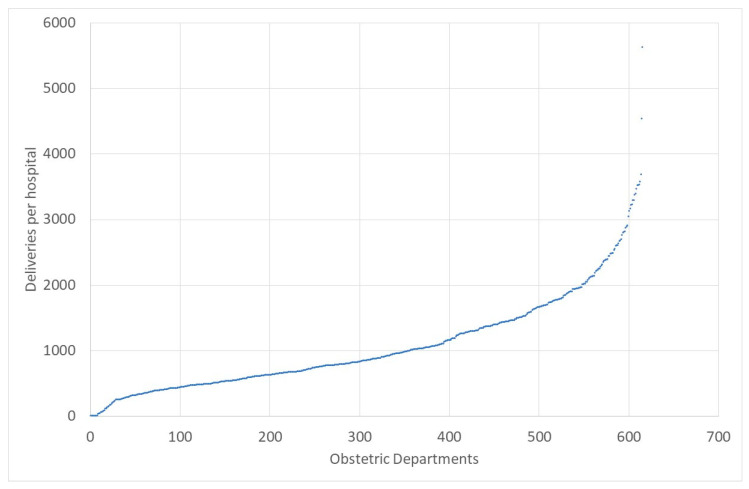
Deliveries per Hospital. Source: own. Data: [11].

**Figure 2 healthcare-12-00009-f002:**
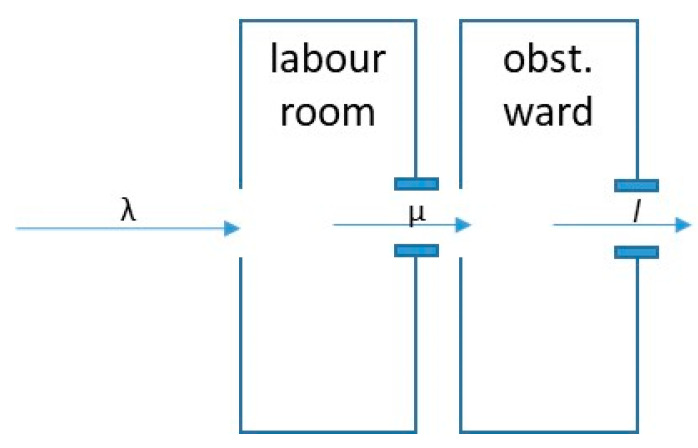
Obstetric Department as a Waiting System. Source: own.

**Figure 3 healthcare-12-00009-f003:**
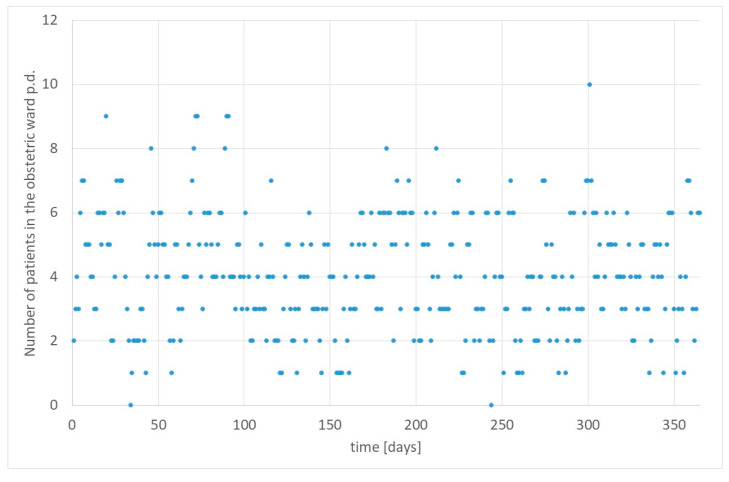
Number of patients in the obstetric ward per day. Source: Own simulation.

**Figure 4 healthcare-12-00009-f004:**
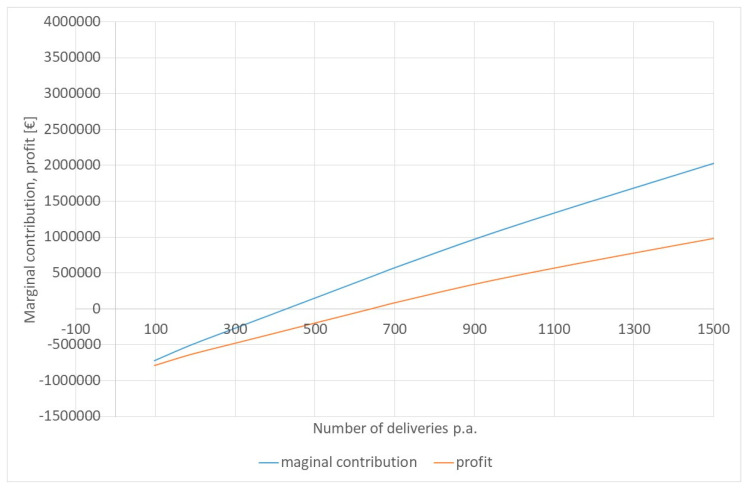
Break-Even Analysis. Source: Own simulation.

**Figure 5 healthcare-12-00009-f005:**
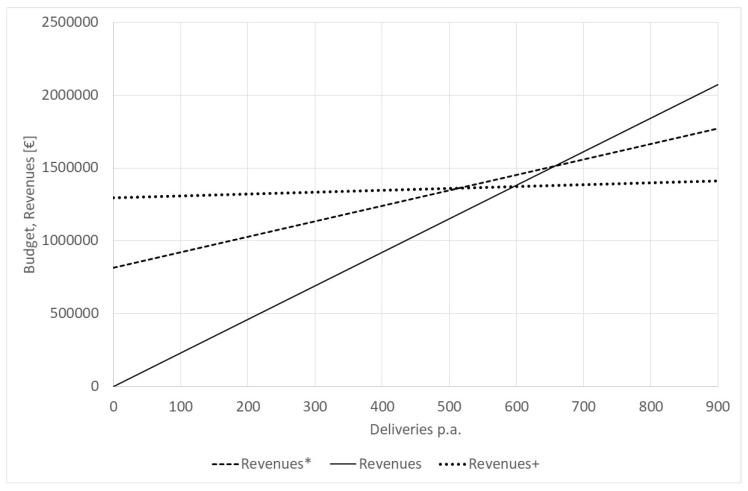
Potential revenue functions of a hospital with 500 deliveries p.a. Source: own.

**Table 1 healthcare-12-00009-t001:** DRG O60D and bed-capacity of German hospitals, 2022. Source: [15].

Number of Hospital Beds	Cases DRG O60A	Cases DRG O60B	Cases DRG O60C	Cases DRG O60D	Total
0–49	0	13	0	396	409
50–99	0	0	0	70	70
100–149	0	122	1094	7897	9113
150–199	14	353	2872	18,207	21,446
200–249	13	462	3447	23,885	27,807
250–299	47	815	5138	28,851	34,851
300–399	137	2163	11,497	59,702	73,499
400–499	172	2149	11,008	56,733	70,062
500–599	157	1633	6576	42,778	51,144
600–799	240	2498	11,705	52,611	67,054
800–999	146	1239	5044	18,131	24,560
>1000	444	3621	13,613	49,271	66,949
Total	1370	15,068	71,994	358,532	446,964

**Table 2 healthcare-12-00009-t002:** Constants of Basic Simulation.

Constant	Parameter	Value	Source
Deliveries per year	*d*	500	parametric constant, assumption
Arrival rate	λ	1.37	λ = *d*/365
Length of stay	*l*	2.9	[12]
	*σ*	0.8	
Nursing cost weight	*c_n_*	0.6702	[12]
Value of nursing cost weight	*v_n_*	230	[12]
Annual salary cost	nurses: *S_n_*	EUR 45,000	collective agreement TV ÖD
	midwife: *S_m_*	EUR 45,000	
	doctor: *S_d_*	EUR 90,000	
Resource utilisation	nurses: *R_n_*	2 h per patient day	[22], assumptions based on [12]
	midwife: *R_m_*	7 h per delivery	
	doctor: *R_d_*	1 h per patient day	
Minimum personnel	nurses: *M_n_*	5 nurses	$\frac{365\cdot 24}{40\cdot 46}$
	midwife: *M_m_*	5 midwives	
	doctor: *M_d_*	5 doctors	
Security level	nurses: *p_n_*	85%	assumption
	midwife: *p_m_*	85%	
	doctor: *p_d_*	85%	
	beds: *p_w_*	85%	
Revenue per delivery	*r*	EUR 2303.53	[12]
variable cost per delivery	*v_d_*	430.94	[12]
allotted overheads per delivery	*v_o_*	697.61	[12]

**Table 3 healthcare-12-00009-t003:** Model Variables. Source: own.

Variable	Explanation
*D_t_*	number of deliveries on day t in labour room
*P_t_*	number of patients on day t in obstetric ward
*X_i_*	number of staff, i = [nurses, midwives, doctors]
*B*	number of beds
*D*	quintile of number of deliveries per day with *p_m_*%
*C*	total cost [EUR]
*R*	total revenues [EUR]
*Π*	profit/loss [EUR]
*m*	marginal contribution [EUR]

**Table 4 healthcare-12-00009-t004:** Cost per Delivery of DRG O60D. Source: [12].

	Cost Medical Personnel	Cost Nursing Personnel	Cost Functional Personnel	Pharmaceuticals	Materials	Overheads	Total
Normal ward	224.88	447.02	32.24	EUR 12.31	EUR 9.63	EUR 399.16	EUR 1125.24
ICU	0.14		0.01	EUR 0.03	EUR 0.02	EUR 0.23	EUR 0.43
Theatre	3.93		3.18	EUR 0.06	EUR 1.33	EUR 3.63	EUR 12.13
Anaesthesia	15.16		10.04	EUR 0.81	EUR 2.62	EUR 6.70	EUR 35.33
Labour ward	189.57		503.08	EUR 24.85	EUR 65.23	EUR 268.35	EUR 1051.08
Endoscopy			0.01				EUR 0.01
Radiology	0.13		0.11		EUR 0.09	EUR 0.12	EUR 0.45
Laboratory	1.68		5.58	EUR 0.73	EUR 4.25	EUR 4.60	EUR 16.84
Diagnostics	3.4	0.17	3.62	EUR 0.03	EUR 0.23	EUR 2.61	EUR 10.06
Therapy	2	0.11	7.17	EUR 0.08	EUR 1.18	EUR 2.72	EUR 13.26
Admission	14.13	0.9	12.01	EUR 0.43	EUR 1.74	EUR 9.49	EUR 38.70
Total	455.02	448.2034	577.05	39.33	86.32	697.61	EUR 2303.53

**Table 5 healthcare-12-00009-t005:** Impact of different security levels. Source: Own simulation.

No. of Deliveries p.a.	Security Level	Total Staff	Beds	Occupancy Rate	Over-Occupancy	Profit
514	50%	15	4	102%	39%	−EUR 177,160.05
	70%	15	5	81%	24%	−EUR 177,160.05
	85%	15	6	68%	8%	−EUR 177,160.05
	95%	15	7	58%	3%	−EUR 177,160.05
	100%	17	10	41%	0%	−EUR 267,160.05
718	50%	15	4	96%	34%	EUR 109,725.84
	70%	15	7	82%	21%	EUR 109,725.84
	85%	15	8	72%	14%	EUR 109,725.84
	95%	17	10	58%	3%	EUR 19,725.84
	100%	21	14	41%	0%	–EUR 160,274.16

**Table 6 healthcare-12-00009-t006:** Impact of resource consumption. Source: Own simulation.

No. of Deliveries p.a.	Change in Time Consumption	Nurses	Midwives	Gynaecologists	Profit
514	−50%	5	5	5	−EUR 177,160.05
	−25%	5	5	5	−EUR 177,160.05
	0%	5	5	5	−EUR 177,160.05
	+25%	5	7	5	−EUR 177,160.05
	+50%	5	9	5	−EUR 267,160.05
718	−50%	5	5	5	EUR 109,725.84
	−25%	5	5	5	EUR 109,725.84
	0%	5	5	5	EUR 109,725.84
	+25%	5	7	5	EUR 19,725.84
	+50%	7	9	5	−EUR 160,274.16
1495	−50%	5	5	5	EUR 1,202,423.58
	−25%	5	7	5	EUR 1,112,423.58
	0	6	9	5	EUR 977,423.58
	25%	9	13	5	EUR 662,423.58
	50%	12	17	5	EUR 347,423.58

## Data Availability

The relevant calculations can be obtained from the author.

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
