# Peer review of "Efficiency of Obstetric Services in Germany—The Role of Variation and Overheads"

_healthcare, 2023, doi:10.3390/healthcare12010009_

Round 1
Reviewer 1 Report
Comments and Suggestions for Authors
The author develops a Monte-Carlo simulation model that analyses the workload of the labour room and the obstetric ward to demonstrate that smaller obstetric departments have disadvantages in comparison to bigger departments.
The simulation model is interesting and suggests that small hospitals cannot break even in this economic model. It gave us some insight into the economies of scale of obstetric departments in these smaller hospitals.
However, the data used for this model consisted mainly of pregnant women who were more than 33 weeks pregnant and whose newborns were delivered vaginally. I have a bit of a question as to whether there is a diagnosis of preterm labor in the delivery of a pregnant woman who is not at full term. In addition, I hope that this model will cover all pregnant women without complications. For pregnant women without complications, they tend to stay in small hospitals for delivery, regardless of the mode of delivery. The inclusion of these people in the model allows for a more comprehensive analysis.
Author Response
Please find attached my review in a separate document.

Reviewer 2 Report
Comments and Suggestions for Authors
See attached review.

Author Response

(The authors gave the same response as above.)

Round 2
Reviewer 2 Report
Comments and Suggestions for Authors
Quick and adequate response to the comments in first round. Congratulations!